# Longer Brace Duration Is Associated with Lower Stress Levels and Better Quality of Life in Adolescents with Idiopathic Scoliosis

**DOI:** 10.3390/children10071120

**Published:** 2023-06-28

**Authors:** Fabrizio Di Maria, Gianluca Testa, Michela Carnazza, Martina Testaì, Vito Pavone

**Affiliations:** 1Department of General Surgery and Medical Surgical Specialties, Section of Orthopaedics and Traumatology, University Hospital Policlinico “Rodolico-San Marco”, University of Catania, 95123 Catania, Italy; gianpavel@hotmail.com (G.T.); michela.carnazza@libero.it (M.C.); vitopavone@hotmail.com (V.P.); 2Unit of Clinical Pediatrics, AOU “Policlinico”, PO “G. Rodolico”, University of Catania, 95123 Catania, Italy; martinat617@gmail.com

**Keywords:** adolescent idiopathic scoliosis, brace treatment, psychological stress, quality of life

## Abstract

Physical and psychological health concerns have been raised due to either spine deformity or orthotic treatment in adolescents with idiopathic scoliosis. To determine whether orthotic bracing duration affects psychological stress and health-related quality of life, a sample of 46 patients (8M, 38F, aged 11–17 years, regularly treated with bracing) with moderate or moderate to severe idiopathic scoliosis were separated into two groups based on whether their treatment duration was up to six months or longer. The brace-related levels of stress and quality of life were investigated in both groups by using the Bad Sobernheim Stress Questionnaire and the Italian Spine Youth Quality of Life, respectively. The questionnaire scores were categorized as low, mean, and high. Our statistical analysis considered the proportion of patients falling into the three categories and the difference in crude score rates between the two groups. Values were considered significant at *p* < 0.05. The proportion of patients with brace-related stress and impaired quality of life was significantly lower in the group treated longer compared to the group that had only received treatment for up to six months (chi-square test, *p* < 0.0001). Overall, mean ± SD BSSQ scored 8.72 ± 4.91 and 12.89 ± 4.65 in group 1 and group 2, respectively (Mann–Whitney *U* test, *p* = 0.008), while ISYQoL scored 19.94 ± 8.21 and 16.07 ± 6.12, respectively. We argue that the differences could depend on both the physical and psychological adaptation patients make to their brace and that more support should be provided to patients when they start to wear their brace.

## 1. Introduction

Adolescent Idiopathic Scoliosis (AIS) is a common disease state characterized by lateral spine curvature with a Cobb angle of more than 10 degrees and rotation of the vertebral column along its longitudinal axis, affecting approximately 1 to 3% of the 10 to 17 year-old subjects [1]. Typically, at the time of diagnosis, patients have a fairly good health status and are usually unaware of their spine deformity.

During the past few decades, studies have found that AIS is positively affected by wearing a rigid thoracolumbar brace, which, acting through external compressive forces, halts or reduces the progression of spine curvature, with longer daily brace wearing associated with greater benefits [2].

Scoliosis per se has the potential to adversely affect many psychological aspects in adolescents and is currently recognized as an important risk factor for psychic discomfort and poor quality of life (QoL). Thus, AIS patients may experience both physical and psychological stress as well as social isolation, depression, and reduced participation in leisure activities [3]. In addition, it has been reported that brace-associated factors may interfere with several aspects of patients’ well-being. Brace external visibility, perceived pain, neurological complications, respiratory discomfort, and emotional distress are among the most common aspects, all of which may lead to variable levels of stress and negatively impact every-day life [4,5,6]. Hence, AIS-related stress levels and poor QoL could be worsened by brace wearing for many hours a day, which in turn may affect patients’ adherence to treatment and lead to disease progression [7]. A number of validated questionnaires specifically targeted to brace-wearing scoliotic populations are available, including the Bad Sobernheim Stress Questionnaire—Brace (BSSQ) and the Italian Spine Youth Quality of Life for “brace” domain (ISYQoL), which have been successfully used in AIS patients to evaluate the short-term effect of brace-related stress level and QoL, respectively [8,9]. It is known that stress associated with brace wearing can deteriorate self-confidence and decrease the activity of AIS patients but it has also been suggested that, after the initial shock of being conscious about the spine deformity and its treatment, the psychological effects of brace wearing tend to decrease [10,11]. However, to the best of our knowledge, whether brace wearing duration has an effect on brace-related levels of stress and QoL has not yet been investigated.

In this study, we first aimed to evaluate whether brace treatment at different time points is associated with stress levels and quality of life in AIS patients with good compliance to bracing after they completed both ISYQoL and the validated Italian version of the BSSQ questionnaire. Secondly, we analyzed whether there is an association between stress level and QoL.

## 2. Materials and Methods

Out of 132 consecutive patients attending the outpatient clinic of the Orthopaedics Unit at the University Hospital Policlinico-San Marco, Catania, Italy, between November 2020 and May 2022, subjects with moderate or moderate to severe AIS were enrolled.

Inclusion criteria were as follows: (1) confirmed diagnosis of moderate to severe AIS (Cobb angle of 21 to 40 degrees for the primary curve) according to SOSORT guidelines [12]; (2) a thoracic or thoracolumbar primary curve; (3) skeletal immaturity with growth cartilage visible on pre-treatment radiographs (Risser score <5); (4) age 10 to 17 years; (5) ongoing brace treatment for at least four weeks. Exclusion criteria were as follows: (1) scoliosis due to other spine disorders or linked to known causes; (2) underlying psychiatric conditions (such as widespread developmental and psychotic disorders). Demographic and relevant clinical data were recorded. The Cobb angle and Risser classification were assessed by the same senior orthopedic surgeon (V.P.) according to current guidelines [12]. To make up for the height loss resulting from spine curvature, individual statures were determined by using the Armspan-to-Height Software [13].

After meeting all the inclusion criteria and accepting the terms of participation, a total of 46 patients (8 males and 38 females, aged 11–16 years) were enrolled in the study. The majority of the 132 patients eligible were excluded for either being too old or having a Cobb angle outside the range of 21–40 degrees or both. Those with a poor or erratic adherence to daily bracing were also excluded. In addition, a small number of eligible AIS patients were not enrolled because they declined the offer to participate in the study. The mean age ± SD of participants was 13.6 ± 1.76 years, and Cobb angle at the beginning of treatment was 27.2 ± 6.24 degrees. All patients were treated with a Sforzesco or Chêneau brace (31 and 15 subjects, respectively), worn daily for at least 14 h per day. The mean ± SD of Risser score was 1.4 ± 1.14, indicating that all patients had skeletal immaturity and were at risk of scoliotic curve progression.

Upon enrollment, subjects were assigned to one of two groups depending on whether their brace duration was less than or equal to six months (group 1; *n* = 14) or longer (group 2; *n* = 32). Vital statistics and other relevant characteristics of enrolled patients are shown separately for each study group in Table 1. Means were analyzed using the Mann–Whitney *U* test and the unpaired *t*-test with unequal variances (Welch’s correction). No difference between groups was found for all vital parameters, Cobb angle, Risser score, and hours of bracing per day. Data were normally distributed within each group, with the exception of age, BMI, and Cobb angle. Female to male ratio was 6:1 for group 1 and 4.33:1 for group 2. Neither the proportion of subjects using a Sforzesco or Chêneau brace nor the number of hours spent in a brace were different between group 1 and group 2.

In both groups, the BSSQ (“brace” version) and ISYQoL (“brace domain” version) questionnaires were used to determine stress levels and quality of life, respectively [9,14].

They were chosen because they are easy to use and widely trusted, as evidenced by their use in a number of previous investigations. Both were completed by patients immediately after having worn their own brace. The selected threshold for statistical significance was at *p* value of less than 0.05.

The ISYQoL measures the health-related quality of life of adolescent with spinal deformity. It consists of 20 items, each scored 0, 1, or 2, with seven items covering the impact of bracing on quality of life. Using the Rasch analysis, the total score is converted into a range of measures expressed on a 0–100% scale (with 100% indicating a very good quality of life). The BSSQ is a short, concise questionnaire consisting of 8 questions that assesses psychological stress induced by wearing a brace. The response to each question is scored on a scale of 0 (most stress) to 3 (least stress), with a maximum achievable score of 24. The subdivision of score values was as follows: 0–8 (high stress), 9–16 (medium stress), and 17–24 (low stress).

The Review Board of the Department of General Surgery and Medical Surgical Specialties (Section of Orthopaedics and Traumatology) was formally informed, and ethical review and approval were waived because all actions in the study protocol were in the framework of routine clinical activity. The study was conducted in accordance with the Declaration of Helsinki. All patients and their parents were informed about the study purpose and design, and informed consent regarding the anonymous publication of pooled data was obtained from at least one parent or legal representative for each study subject.

Data are presented as means plus or minus standard deviation (SD) or percentage, as appropriate. Normal distribution of data was ascertained by the D’Agostino–Pearson normality test. Chi-squared statistical analysis was used to compare the percentages of patients with different score categories regarding stress levels and QoL in group 1 and group 2. The Mann–Whitney *U* test was used to analyze questionnaire score differences between the two groups. The association between BSSQ and ISYQoL scores was explored by the Spearman rank correlation coefficient test. All statistical analyses were performed using the GraphPad Prism Version 5.0 computing package (GraphPad Software Inc., San Diego, CA, USA).

### Validation of the Italian Version of the BSSQ Questionnaire

A validated Italian BSSQ questionnaire, translated from a Spanish validated version of the questionnaire, was used [15]. Validation was completed in a sample of 18 AIS patients (16F and 2M, aged 11–16 years), with a mean ± SD Cobb angle of 26.10 ± 6.61 degrees at the time of questionnaire self-administration. The process of validation involved two stages: first, a translation and a back translation were made to develop the Italian version; subsequently, all subjects answered the questionnaire twice at a 5 days interval. Internal consistency and the reliability of questionnaire items were determined through the calculation of Cronbach’s alpha coefficient and the Cohen’s kappa coefficient, respectively. The overall Cronbach’s alpha and Cohen’s kappa values were 0.76 and 0.73, respectively, indicating a satisfactory internal consistency and good agreement between responses to the Italian BSSQ. The Italian version of the BSSQ questionnaire is available in the Appendix A.

## 3. Results

Overall, the mean ± SD of BSSQ and ISYQoL questionnaire scores were 11.26 ± 5.13 and 17.59 ± 7.18, respectively. A slightly higher stress level and lower QoL was found in females. Using cut-off scores from the BSSQ questionnaire [12], patients turned out to have low, medium, and high stress levels in 17%, 48%, and 35%, respectively.

Significantly lower brace-related stress, but not better quality of life, was observed in group 2 compared to group 1. Mean ± SD BSSQ scored 8.72 ± 4.91 and 12.89 ± 4.65 in group 1 and group 2, respectively, while ISYQoL scored 19.94 ± 8.21 and 16.07 ± 6.12 (Table 2).

At the same time, the proportion of patients with both mean and high stress levels was significantly lower for group 2 (longer brace treatment duration) compared to group 1 (shorter brace treatment duration; *p* < 0.0001, Figure 1A). Consistently, QoL scored better in a significantly higher proportion of patients in group 2 compared to group 1 (*p* < 0.0001, Figure 1B).

Finally, no significant association was found between BSSQ and ISYQoL questionnaire rates (Spearman correlation coefficient was r = −0.221; *p* = 0.14; Figure 2).

## 4. Discussion

Although the exact types and dimensions of AIS-induced psychological effects are uncertain, a large body of evidence indicates that increased stress levels and impaired QoL are among the most frequent. These negative outcomes are not only associated with physical deformity but are also increased by treatment based on long-term brace wearing for many hours per day.

In previous studies, we have shown that adolescents with moderate or moderate to severe idiopathic scoliosis and ongoing conservative brace treatment have variable stress levels and a poor quality of life, as assessed by the brace-domain of the BSSQ and ISYQoL questionnaires, respectively. In this study, we aimed to evaluate whether ongoing brace treatment at different time points in AIS patients with good compliance to bracing is associated with different levels of psychological stress and impaired quality of life.

Interestingly, our results from the present study indicate that a larger proportion of patients that have received treatment for longer than six months have lower levels of psychological stress and better QoL compared to those treated for at least 4 weeks but less than 6 months, thus suggesting that, possibly due to self-motivation and good compliance, patients become accustomed to the orthotic treatment and are therefore more equipped to adapt to and deal with its negative aspects. The design of our study did not allow us to ascertain whether psychological or physical factors or both are the major determinants of adaptation in our patients. This distinction is difficult to determine because many components may contribute to the complexity of psychological concerns. Scoliosis in adolescents occurs at a time when self-image and social relationships are becoming increasingly important; thus, it could have a negative impact on a variety of psychological aspects and may be a source of fear and lack of self-esteem, which in turn may cause stress, poor QoL, and social embarrassment [16]. The major negative impacts in the early period of bracing could also be attributed to the fact that patients will have only recently been made aware of their diagnosis, which may cause discomfort due to their increase awareness of both the disease and the burden of treatment.

Brace wearing for many hours is the cornerstone of non-surgical treatment but its efficacy is strictly related to treatment compliance [17]. Good compliance is of pivotal importance to reduce the hazard of curve progression and avoid the need for surgery [2]. Previous studies have shown that brace wearing impacts patients’ mental well-being and negatively affects their QoL [8,18,19]. Overall, AIS may amplify stress and adversely affect QoL. Our findings suggest that this is particularly evident in the early months of brace treatment; however, the negative effects seemingly decrease if treatment continues uninterrupted.

Over the past two decades, several tools have been proposed for the evaluation of stress and QoL in AIS patients. The SRS-22 and BrQ questionnaires have been extensively used [20,21]. However, these tools are mostly focused on evaluating QoL rather than stress levels. In this study, two easy-to-use, self-completed questionnaires (the brace oriented BSSQ and ISYQoL) were used to assess stress levels and QoL, respectively, and the putative association between them was also explored. After using the BSSQ (brace) questionnaire, a recent study shown a direct relationship between the number of bracing hours per day and stress levels [22]. However, in contrast with our results, the authors of this study did not find any correlation between the length of bracing in months and the BSSQ score.

Similarly to other studies [10,23,24], we also noticed a slightly higher stress level and lower QoL in females than males (data not shown). This is because teenage girls, compared to boys, are maybe more concerned with body self-image and other cosmetic issues.

As a secondary endpoint, we have shown that the Italian version of the BSSQ (brace) questionnaire can be used reliably to assess brace-related stress levels in Italian adolescents with idiopathic scoliosis. Previous studies have found a significant relationship between SRS-22, BSSQ, and BrQ tools [11,24]. In contrast, although both stress levels and QoL both appear to be related to the duration of brace treatment, no association was found between BSSQ and ISYQoL in our study. This result could be a consequence of the questionnaires’ different structures and purposes.

In order to assess how the length of bracing period during adolescence may influence stress level and QoL, we arbitrarily divided subjects into two groups: one group had experience brace treatment for less than or equal to six months, whereas the other group had been treated continuously for more than six months. Within the former group, compared to the latter, a significantly higher proportion of subjects had higher stress levels and lower QoL. These results are consistent with a previous study that showed that 84 percent of parents described the initial bracing period as stressful, especially due to the fact that it affected their participation in daily leisure activities, such as sports and social events [3]. It is possible that the negative psychological aspects associated with brace treatment could induce fear in patients as well as a decrease in self-confidence, especially when they have just learned of their diagnosis and its implications, including the recommended long bracing treatment period.

We recognize that the findings in this study are limited due to the cross-sectional nonrandomized design. Other study limitations include the small sample size and unequal gender distribution, as well as the unbalanced brace model used by our patients.

## 5. Conclusions

In conclusion, among AIS patients, increased psychological stress and poor QoL induced by brace wearing are particularly evident during the early bracing period, although both improve significantly over time (with increasing the brace treatment duration). Whether or not this improvement is attributable to patients adapting to brace treatment or patient tolerance is arguable. It is possible that other factors, including slowing curve progression, self-motivation, education level, and socioeconomic status, play a role. However, it is also possible that the negative effects of bracing patients experience in the initial stages of treatment could lower or even halt the patients’ adherence to treatment, potentially leading to spine deformity progression. The results of this study indicate that AIS patients in the initial stages of conservative scoliosis treatment are more susceptible to the negative effects of brace wearing. Hence, early intervention strategies, including the provision of psychological support, are advisable.

## Figures and Tables

**Figure 1 children-10-01120-f001:**
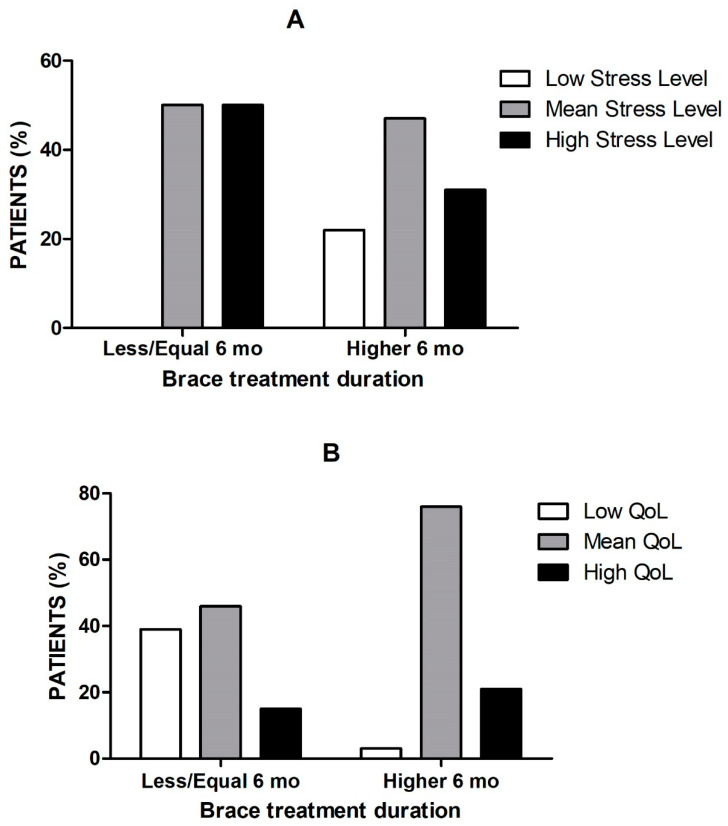
Frequency distribution of low, mean, and high rate of (**A**) stress level and (**B**) quality of life according to the duration of bracing treatment. In patients with a brace treatment duration longer than 6 months (group 2; right columns), the mean and high rates of stress level are lower, whereas those of quality of life are higher compared to those with a treatment duration of up to or lower than 6 months (group 1; left columns)—chi-square test, *p* < 0.0001.

**Figure 2 children-10-01120-f002:**
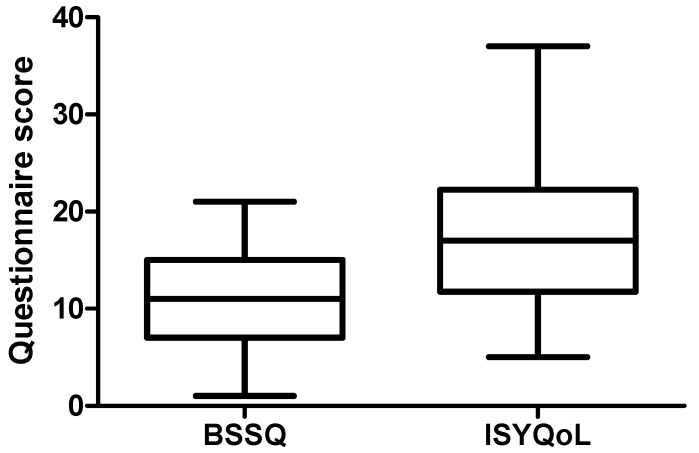
Box-and-whiskers plot of BSSQ (mean 11.26 ± 5.13; 95% CI 9.74–12.78) and ISYQoL (mean 17.59 ± 7.18; 95% CI 15.45–19.72) overall scores in the whole study population (n = 46). No significant correlation between BSSQ and ISYQoL was found by using the non-parametric Spearman test (r = −0.221 (95% CI −0.49–0.08) *p* = 0.14, NS).

**Table 1 children-10-01120-t001:** Vital statistics, bracing hours, and months of bracing treatment in patients of group 1 and group 2 with “short” and “long” bracing treatment duration, respectively. Mean ± SD, minimum and maximum individual values, and *p*-values obtained by analyzing the data by using both the Mann–Whitney test and the unpaired *t*-test with unequal variances are shown. The “null” hypothesis that the two groups have equal means was true, except for the bracing treatment duration. (BMI = Body Mass Index).

	Group 1 “Short Duration”		Group 2 “Long Duration”		Mann–Whitney *U* Test	Unequal Variances *t*-Test
	Mean ± SD	Min–Max	Mean ± SD	Min–Max	*p*-Value	*p*-Value
Age (years)	13.36 ± 1.45	11–16	13.63 ± 1.84	11–16	0.6360	0.6002
Height (cm)	157.7 ± 10.22	136–170	154.9 ± 12.50	132–178	0.4585	0.4252
Weight (kg)	49.21 ± 8.84	31–65	48.10 ± 8.52	30–68	0.6933	0.6928
BMI (kg/m^2^)	19.66 ± 2.23	16.76–25.39	19.94 ± 1.72	16.00–23.31	0.4238	0.6857
Cobb angle (degrees)	26.57 ± 4.91	21–40	27.13 ± 5.87	21–40	0.9617	0.7432
Risser score (units)	1.357 ± 1.08	0–3	1.344 ± 1.18	0–4	0.9114	0.9703
Bracing (hrs/day)	16.29 ± 1.44	14–18	16.06 ± 1.43	14–20	0.4512	0.6322
Treatment duration (months)	2.43 ± 1.95	1–6	17.84 ± 10.15	8–46	<0.0001	<0.0001

**Table 2 children-10-01120-t002:** Mean values ± SD and 95% confidence intervals of BSSQ and ISYQoL scores in AIS patients with ongoing brace treatment up to six months (group 1; n = 18) or higher (group 2; n = 28). Between-group comparison was made by using the Mann–Whitney *U* test.

	Group 1	Group 2	Mann–Whitney Test
BSSQ	8.72 ± 4.91 (95% CI 6.28–11.16)	12.89 ± 4.65 (95% CI 11.09–14.70)	*p* = 0.008
ISYQoL	19.94 ± 8.21 (95% CI 15.86–24.03)	16.07 ± 6.12 (95% CI 13.70–18.45)	NS

## Data Availability

The data presented in this study are available on request from the corresponding author. The data are not publicly available due to privacy restriction.

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
