# Peer review of "Longer Brace Duration Is Associated with Lower Stress Levels and Better Quality of Life in Adolescents with Idiopathic Scoliosis"

_children, 2023, doi:10.3390/children10071120_

Round 1
Reviewer 1 Report
The main issue concerns methods. Indeed, there is no clear accounting for confounders / covariates. Maybe the age, sex, type of brace, Cobb angle, BMI, social level, cultural level etc. etc. influence more than duration of brace treatment. For this, you should match the subjects of the 2 groups according with age etc, perform univariate analysis for all covariates between groups (e.g. age of group 1 vs age of group 2, sex of group 1 vs sex of groups 2 etc), and perform multivariate analysis. I suggest reading the following article:
https://www.ncbi.nlm.nih.gov/pmc/articles/PMC4756459/#:~:text=A%20matched%20cohort%20study%20involves,matched%20on%20certain%20baseline%20characteristics.
Moreover, there is no clear flow chart on inclusions. Why did you include only 46 patients out of 132? Lack of data, refusals, other diagnosis? Please add a flow chart, e.g. 132 eligible, 10 excluded for other diagnosis, 20 excluded for lacking data, 12 excluded for psychiatric conditions, 9 refusals from parents etc.
Author Response
"The main issue concerns methods. Indeed, there is no clear accounting for confounders / covariates. Maybe the age, sex, type of brace, Cobb angle, BMI, social level, cultural level etc. etc. influence more than duration of brace treatment. For this, you should match the subjects of the 2 groups according with age etc, perform univariate analysis for all covariates between groups (e.g. age of group 1 vs age of group 2, sex of group 1 vs sex of groups 2 etc), and perform multivariate analysis. I suggest reading the following article:
https://www.ncbi.nlm.nih.gov/pmc/articles/PMC4756459/#:~:text=A%20matched%20cohort%20study%20involves,matched%20on%20certain%20baseline%20characteristics "
Thank you for your helpful suggestion, and article link. We fully agree with you but before proceeding on to analysis of covariance we calculated the vital statistics data separately for the individual subjects of group1 and group2 and reported the results in the new version of Table 1 below. Hopefully, accounting for confounders / covariates is now clear. Table 1 will be included and issue mentioned in the revised manuscript. (Table attached here)
"Moreover, there is no clear flow chart on inclusions. Why did you include only 46 patients out of 132? Lack of data, refusals, other diagnosis? Please add a flow chart, e.g. 132 eligible, 10 excluded for other diagnosis, 20 excluded for lacking data, 12 excluded for psychiatric conditions, 9 refusals from parents etc. "
We do not believe that adding a flow chart would help that much. The selection / inclusion procedure is clearly explained by the strict enforcement of criteria listed (see page 2, lines 67-75). Only patients fully complying with these criteria were enrolled the majority of 132 were excluded for having age and Cobb angle outside the range of 10-17 yrs and 21-40 degrees, respectively or both, and also for having a poor adherence to bracing. In addition, a small number of eligible AIS patients were not enrolled for denial to participate in the study. Text was changed accordingly.

Reviewer 2 Report
Dear Author,
Thank you for the opportunity to review this article.
The introduction is relevant regarding to the field of interest. Indeed bracing for scoliosis may be a source of distress for the pediatric patients. The hypothesis is whether the levels of distress are age-related.
In comparison, scoliosis surgery for higher values of Cobb Angle drastically influence QoL, and more information can be found here https://www.mdpi.com/1648-9144/58/5/674 that could be added in the Introduction. A comparison between conservative treatment QoL and surgically treated patients QoL can be assesed.
Materials and Methods:
Are there any reasons you chose only two types of braces? What were your indications for either one? Did all the scoliotic patients have similar curvatures? Do you consider them to be superior to Boston/SpineCor/LionArtBrace? Please comment on those issues.
Results
In you paper, you stated that the patients worn the brace for at least 14 hours. There are a lot of studies showing that curvature progression is dependent on the number of hours spent by bracing. It would be helpful to include a table in which you compare QoL with number of hours spent in brace. Have the patients that spended less hours in bracing had a better QoL?
QoL is important, but the Cobb angle correction in time is also important. Please provide a table with this data. If the patient wore the brace for a smaller amount of time, having a bigger QoL, but also a greater curvature, than the QoL should be refined.
Discussions
Did any child benefit from Psychological support? Some authors stated that it should be a standard in scoliosis management that could greatly improve compliance.
Why only include patients with less than 40 degree Cobb angle? Do you operate on patients with Cobb angles above 40 degrees?
Did any vital statistics correlate with the stress scores? Or with the curvature progression?
Conclusions:
Conclusions should be limited to 3 main ideas.
Discussion parts should be moved to Discussions (study limitations, unanswered questions, etc.).
Author Response
Dear Reviewer,
thank you for your comments.
"The introduction is relevant regarding to the field of interest. Indeed bracing for scoliosis may be a source of distress for the pediatric patients. The hypothesis is whether the levels of distress are age-related."
The hypothesis is whether longer bracing is associated to lower psychological distress and better QoL.
In comparison, scoliosis surgery for higher values of Cobb Angle drastically influence QoL, and more information can be found here:
https://www.mdpi.com/1648-9144/58/5/674
that could be added in the Introduction. A comparison between conservative treatment QoL and surgically treated patients QoL can be assesed.
Thank you for your comment. We took it into careful consideration and although current evidence clearly indicates that surgery improves QoL of patients with severe scoliosis, we didn’t mention it because is not strictly relevant for the purpose of our study focused on patients with moderate or moderate to severe AIS under orthotic brace treatment. To give readers the clearest picture, it is sufficient that the introduction is logically organized on topics leading to the question posed in the paper.
Are there any reasons you chose only two types of braces? What were your indications for either one? Did all the scoliotic patients have similar curvatures? Do you consider them to be superior to Boston/SpineCor/LionArtBrace? Please comment on those issues
Neither we did choose brace types in our study nor we did consider them to be superior to other models. Probably due to market reasons, they are just the most used in Italy, and they came out as it was during the recruitment process. In addition, their proportions were not significantly different between groups (Fisher’s exact test; p=0.208, NS). Please, see table attached here.
In your paper, you stated that the patients worn the brace for at least 14 hours. There are a lot of studies showing that curvature progression is dependent on the number of hours spent by bracing. It would be helpful to include a table in which you compare QoL with number of hours spent in brace. Have the patients that spended less hours in bracing had a better QoL?
A minimum bracing time of 14 hours per day was accepted as an inclusion criterion – and checked upon enrollment – to warrant good compliance (and avoid bias) in our population sample. On average subjects in both groups weared their brace for more than 16 hours per day. Hence, the design of our study as well as the bracing time homogeneity does not allow us to assess correlation between bracing hours and both curve progression and QoL.
QoL is important, but the Cobb angle correction in time is also important. Please provide a table with this data. If the patient wore the brace for a smaller amount of time, having a bigger QoL, but also a greater curvature, than the QoL should be refined.
Assessment of Cobb angle correction related to the duration of brace wearing does not comply with the adopted study protocol and is beyond the purpose of our study. This question point is similar to the previous one. We apologize, but our answer must be also similar.
Did any child benefit from Psychological support? Some authors stated that it should be a standard in scoliosis management that could greatly improve compliance.
No psychological support was provided. Subjects upon enrollment only received recommendation to wear their brace for at least 16 hours per day as part of the routine clinical practice.
Why only include patients with less than 40 degree Cobb angle? Do you operate on patients with Cobb angles above 40 degrees?
The narrow range was chosen to ensure population sample homogeneity. Moreover, the classification based on Cobb angle define moderate and moderate-to-severe curve from 21° to 40°. Above this range, we usually still continue conservative treatment but indications could change between patients.
Did any vital statistics correlate with the stress scores? Or with the curvature progression?
Vital statistics falls into the domain of descriptive data and usually is not the object of statistical comparison to the study dependent outcomes. Rather, it may be important to avoid bias by ensuring that group data are not different. In the revised manuscript, this aspect has been taken into account by calculating and displaying separately putative covariates (e.g age, gender, BMI, Cobb angle etc.) by study groups (new Table 1 attached).
Conclusions should be limited to 3 main ideas.
Discussion parts should be moved to Discussions (study limitations, unanswered questions, etc.).
Thanks for your valuable suggestion. Although it sounds more likely as an editorial indication, we moved unanswered questions and study limitations to the discussion section. We also limited conclusions.

Reviewer 3 Report
The purpose of this study was to investigate the differences in stress and quality of life according to the period of wearing braces in adolescents with Idiopathic scoliosis. The results showed that the brace wearing group for less than 6 months had higher stress and lower quality of life than the brace wearing group for more than 6 months. The subjects were divided into less than 6 months and more than 6 months according to the period of wearing the brace, but the validity of the study results could not be secured because the data for the equivalence test of the two groups were not presented. For example, it is a question of whether the two groups have the same degree of disease, gender, age, etc. In addition to this, the results of this study can be trusted only when the variables that can threaten the validity of the study results are reviewed first. As the researcher stated in the introduction, previous studies have reported that wearing assistive devices causes stress and lowers the quality of life. Although the researcher described this study as the first study, studies related to the wearing of brace report the effect of the wearing period of brace. Researchers need to find and describe the logic and explanatory grounds that can support the results of this study through review of previous studies.
Additionally, in the introduction, the reasons for the need to find out the difference in stress and quality of life according to the wearing period of braces should be presented in detail. In addition, the reliability and validity of the tools used for measurement in the research method must be reported. The presentation of tables should be improved. Table 1 should present the comparative results by group.
Author Response
The purpose of this study was to investigate the differences in stress and quality of life according to the period of wearing braces in adolescents with Idiopathic scoliosis. The results showed that the brace wearing group for less than 6 months had higher stress and lower quality of life than the brace wearing group for more than 6 months. The subjects were divided into less than 6 months and more than 6 months according to the period of wearing the brace, but the validity of the study results could not be secured because the data for the equivalence test of the two groups were not presented. For example, it is a question of whether the two groups have the same degree of disease, gender, age, etc. In addition to this, the results of this study can be trusted only when the variables that can threaten the validity of the study results are reviewed first. As the researcher stated in the introduction, previous studies have reported that wearing assistive devices causes stress and lowers the quality of life. Although the researcher described this study as the first study, studies related to the wearing of brace report the effect of the wearing period of brace. Researchers need to find and describe the logic and explanatory grounds that can support the results of this study through review of previous studies.
You are perfectly right. We calculated the vital statistics data separately for the individual subjects of group1 and group2 and reported the results in a new version of Table 1 to be included in the manuscript. Figures in group1 and group2 are exactly superimposable and the only significant difference found between the two groups was for treatment duration (p<0.0001). Please, see table 1 attached.
Additionally, in the introduction, the reasons for the need to find out the difference in stress and quality of life according to the wearing period of braces should be presented in detail. In addition, the reliability and validity of the tools used for measurement in the research method must be reported. The presentation of tables should be improved. Table 1 should present the comparative results by group.
To address the “reasons for the need to find out the difference in stress and quality of life according to the wearing period of braces”, we added the following statements in the introduction (page 2, line 56):
-It is known that stress associated to brace wearing can deteriorate self-confidence and decrease activity of AIS patients but it has also been suggested that after the initial shock of being conscious about the spine deformity and its treatment, the psychological effects of brace-wearing experience tend to decrease.-

Reviewer 4 Report
This is an interesting ehich suggest it is important to support patients espescially starting period of brace treatment in AIS. If the stress decreases, patient may wear the brace longer time. I am not sure which is beneficial or not.
Author Response
Dear Reviewer,
thank you for your comments.
Round 2
Reviewer 1 Report
Thanks for the revisions.
There is still a problem of method because many confounders are not assessed and inclusions are few.
Could you rephrase the sentence ""Upon enrollment subjects were divided into two groups according to the duration of bracing of less than or equal to six months " ? I'm not sure I understand the study design. Is it a "quasi" prospective cohort without calculation of the number of subjects required or a retrospective analysis of a cohort included prospectively? Were the patients "enrolled" when prescribing the brace with assignment to a "long-lasting" group or a "short-lasting" group?
In addition, I suggest defining "short" and "long" groups instead of group 1 and 2, and writing exact p-values of all tests in table 1, you should present with groups and p-values in horizontal and the list of variates in vertical, as follows
Group "short" Group "long" p-value
Age Mean, SD, min, max Mean, SD, min, max ...............
Sex
Risser sign
etc
etc
Author Response
Thank you for your comments.
"There is still a problem of method because many confounders are not assessed and inclusions are few."
It is quite unusual in the reviewing process that authors are called to respond a new second set of questions from the same reviewer. But since we like fair jobs and want to improve our manuscript before publication, we willingly do our best to respond.
By the Mann-Whitney U test we found that means and SDs of major confounders/covariates were overlapping and not statistically different between the two groups. To be sure of having coped with the possibility that our groups had unequal variances, the “null” hypothesis that the two populations had equal means was also tested by the two-tailed Welch’s t-test (see table 1). This is a quite unusual but robust and reliable statistical test proving that the two groups have the same degree of disease (Cobb), gender, age etc.
We strongly believe that the overlapping means and SD and high p-values are sufficient to conclude that covariates could have no or negligible influence on study results. We accept your point that two groups of pair-matched patients would have been better, but our patient population was not suitable for such a design. Furthermore, going back to your previous observation (and the article you suggested us to read), we must point out that the Cox regression also called “proportional hazards regression” model is better applied for investigating the effect of several variables upon the time a specified event takes to happen, which, clearly is not the purpose of our study.
"Could you rephrase the sentence "Upon enrollment subjects were divided into two groups according to the duration of bracing of less than or equal to six months"? I'm not sure I understand the study design. Is it a "quasi" prospective cohort without calculation of the number of subjects required or a retrospective analysis of a cohort included prospectively? Were the patients "enrolled" when prescribing the brace with assignment to a "long-lasting" group or a "short-lasting" group?"
Patients were already on bracing from not less than 4 weeks when enrolled in the study. The study was neither prospective nor retrospective (see inclusion criteria on page 2). Questionnaires were completed cross-sectionally. We refrased the sentence as follows:
“Upon enrollment subjects were assigned to one of two groups according to the duration of their ongoing bracing treatment of less than or equal to six months (group 1; n = 14) or longer (group 2; n = 32).” …
All AIS patients were already known to be under brace treatment and regularly followed up in our outpatient clinic. After having checked they were suitable for enrollment throughout clinical records, they and their parent(s) were thoroughly informed about the study purpose and protocol, consisting essentially of cross-sectional filling of BSSQ and ISYQoL questionnaires as an add-on of the routine clinical activity.
"In addition, I suggest defining "short" and "long" groups instead of group 1 and 2, and writing exact p-values of all tests in table 1, you should present with groups and p-values in horizontal and the list of variates in vertical, as follows
Group "short" Group "long" p-value
Age Mean, SD, min, max Mean, SD, min, max ...............
Sex
Risser sign
Etc"
“Group 1 and 2” is clear enough and we modified the group heading only in Table 1. We also rearrange the table according to your suggestion. Table attached.
Results show that baseline characteristics of the patients are balanced between the groups and that adjustments are not needed

Reviewer 2 Report
Ok.
Author Response
Thank you
Reviewer 3 Report
Thank you for your effort to revise the manuscript.
I cannot find response below my previous comments.
As the researcher stated in the introduction, previous studies have reported that wearing assistive devices causes stress and lowers the quality of life. Although the researcher described this study as the first study, studies related to the wearing of brace report the effect of the wearing period of brace. Researchers need to find and describe the logic and explanatory grounds that can support the results of this study through review of previous studies.
There was insufficient about measure of the ISYQoL. Please, refer to my previous comments.
In addition, the reliability and validity of the tools used for measurement in the research method must be reported.
Author Response
Thank you for your comments.
"I cannot find response below my previous comments."
We apologize, but your previous comments were rather vague or low-specific. Therefore, we thought they were just comments rather than questions asking for answers or clarification.
"As the researcher stated in the introduction, previous studies have reported that wearing assistive devices causes stress and lowers the quality of life. Although the researcher described this study as the first study, studies related to the wearing of brace report the effect of the wearing period of brace. Researchers need to find and describe the logic and explanatory grounds that can support the results of this study through review of previous studies."
Either we don’t grasp the reviewer’s point or the above sentence is blurry. We do understand that the reviewer says there is some inconsistency in our statements (it’s that correct?). As far as we know, the only study reporting the effect of brace wearing time (either hours or months) and stress level assessed by the BSSQ is that of Behdarvandan et al. ( 20.). Unfortunately, this recently published article is written in Iranian (abstract in English, but not fully comprehensive). We asked the first author to explain their findings, and he (or she) substantially answered what we reported on page 7, lines 224-227) with the following paragraph: “One recent study has shown direct relationship between the number of bracing hours per day and stress level assessed by BSSQ (brace) questionnaire [20]. In contrast with our results, the same authors, however, did not find any correlation between the length of bracing in months and the BSSQ score.”
We also modified the sentence on page 2, lines 59-61 as follows: “To the best of our knowledge, however, the brace-related time point levels of stress and QoL by comparing directly patients with different brace wearing duration have not been investigated so far.”
"There was insufficient about measure of the ISYQoL. Please, refer to my previous comments."
You didn’t even mention ISYQoL in your previous comments. (?)
"In addition, the reliability and validity of the tools used for measurement in the research method must be reported."
The reliability and validity of BSSQ and ISYQoL are well known and largely accepted by the scientific community. We have justified their use at page 3, lines 109-113.
Round 3
Reviewer 1 Report
The most important issues have not been completely addressed, e.g. many potential confounders have not been assessed.
Moreover, the Risser sign of short duration group is very low, whereas patients with Risser 0 /5 usually require long duration (years).
Reviewer 3 Report
Thank you for your response and revising your manuscript.